# In Vitro Inhibitory Effects of Polyphenols from *Flos sophorae immaturus* on α-Glucosidase: Action Mechanism, Isothermal Titration Calorimetry and Molecular Docking Analysis

**DOI:** 10.3390/foods12040715

**Published:** 2023-02-07

**Authors:** Yuhong Gong, Jun Li, Jinwei Li, Li Wang, Liuping Fan

**Affiliations:** 1State Key Laboratory of Food Science & Technology, Jiangnan University, 1800 Lihu Avenue, Wuxi 214122, China; 2School of Food Science and Technology, Jiangnan University, 1800 Lihu Avenue, Wuxi 214122, China; 3Collaborat Innovata Ctr Food Safety & Qual Control, Jiangnan University, 1800 Lihu Avenue, Wuxi 214122, China

**Keywords:** *Flos sophorae immaturus*, polyphenols, α-glucosidase, circular dichroism, isothermal titration calorimetry, molecular docking

## Abstract

*Flos sophorae immaturus* (FSI) is considered to be a natural hypoglycemic product with the potential for a-glucosidase inhibitory activity. In this work, the polyphenols with α-glucosidase inhibition in FSI were identified, and then their potential mechanisms were investigated by omission assay, interaction, type of inhibition, fluorescence spectroscopy, circular dichroism, isothermal titration calorimetry and molecular docking analysis. The results showed that five polyphenols, namely rutin, quercetin, hyperoside, quercitrin and kaempferol, were identified as a-glucosidase inhibitors with IC_50_ values of 57, 0.21, 12.77, 25.37 and 0.55 mg/mL, respectively. Quercetin plays a considerable a-glucosidase inhibition role in FSI. Furthermore, the combination of quercetin with kaempferol generated a subadditive effect, and the combination of quercetin with rutin, hyperoside and quercitrin exhibited an interference effect. The results of inhibition kinetics, fluorescence spectroscopy, isothermal titration calorimetry and molecular docking analysis showed that the five polyphenols were mixed inhibitors and significantly burst the fluorescence intensity of α-glucosidase. Moreover, the isothermal titration calorimetry and molecular docking analysis showed that the binding to α-glucosidase was a spontaneous heat-trapping process, with hydrophobic interactions and hydrogen bonding being the key drivers. In general, rutin, quercetin, hyperoside, quercitrin and kaempferol in FSI are potential α-glucosidase inhibitors.

## 1. Introduction

*Flos sophorae immaturus* (FSI) originated in Asia and is widely cultivated in Asia, including China, South Korea, Japan, etc. FSI exhibited hypoglycemia, and antibacterial and antioxidant properties [1]. These functions of FSI are mainly attributed to various phytochemicals, including rutin and quercetin [2]. It is particularly worth mentioning that the content of rutin in FSI is the highest in the plant kingdom, with rutin content up to 20% of the total mass of FSI, so FSI is a major source of rutin and its aglycone derivative [3]. In addition, a previous study reported that FSI possessed hypoglycemic effects [4]. However, the hypoglycemic mechanism of FSI is unclear.

α-glucosidase is a key enzyme in the hydrolysis of carbohydrates to glucose [5]. Therefore, exploring substances with significantly α-glucosidase inhibitory activity is important for hypoglycemia. Polyphenols and tannins in natural products were confirmed to have the potential ability to inhibit α-glucosidase activity. Cao et al. [6] found that the tannins effectively inhibit intestinal a-glucosidase activity with Ki values in the same range as those of synthetic inhibitors (acarbose), which are widely used to treat type 2 diabetes. Chen et al. [7] found that quercetin and kaempferol-monoglycoside-based flavonoid glycosides in *Moringa oleifera Lam.* might be the main bioactive components for its a-glucosidase inhibit activity. Bao et al. [8] found that niazirin could improve a-glucosidase inhibition activity and hyperglycemia. Cardullo et al. [9] found that valoneic acid dilactone, three tetragalloyl glucose isomers and 1,2,3,4,6-penta-Ogalloyl-β-glucose (from *Castanea sativa wood*) has higher a-glucosidase inhibition ability. Fang et al. [10] found that astilbin, morin and naringenin (from *Lithocarpus polystachyus Rehd*) have the highest a-glucosidase inhibition abilities, and a higher value of natural α-glucosidase inhibitors compared to established drugs [11].

Previous studies have shown that FSI contains a variety of polyphenolic substances, including rutin, chrysin, kaempferol, quercetin, isorhamnetin, hyperoside and quercitrin [12]. However, the different polyphenols show mutual effects in terms of enzyme inhibition activity, etc. The main components of FSI with α-glucosidase inhibitory activity have not yet been validated and the mechanism of inhibition is unclear. Therefore, the identification of the different polyphenols in FSI and the interaction on the α-glucosidase inhibitory activity need to be further explored.

Therefore, this study was conducted to identify the major polyphenols in FSI, then compare the α-glucosidase inhibitory effects of these polyphenols. Furthermore, the inhibitory mechanism of FSI polyphenols on α-glucosidase was preliminarily elucidated by IC_50_ assay, the omission method, interaction, type of inhibition, fluorescence spectroscopy, circular dichroism (CD), isothermal titration calorimetry (ITC) and molecular docking.

## 2. Materials and Methods

### 2.1. Materials and Reagents

Dried FSI (sun dried with the water content of 8.0 ± 0.5%) was purchased from Hebei Anguo Yao Yuan Trading Co., Ltd. (Baoding, China). Rutin (RU), quercetin (QU), hyperoside (HY), quercitrin (QI), kaempferol (KA), gentianoside, protocatechuic acid, isorhamnetin, kaempferol-3-O-rutinoside, narcissoside, chlorogenic acid, 4-nitrophenyl β-D-glucopyranoside (PNPG), acarbose, α-glucosidase (Yeast) (100 U/mg) and all chemicals were obtained from Shanghai Yuan Ye Biological Technology Co., Ltd. (Shanghai, China).

### 2.2. Preparation of the FSI Extract

Dried FSI (1.25 g) was extracted with 40 mL 70% (*v*/*v*) ethanol on a magnetic stirrer (C-MAG HS 7, IKA, Staufen, Germany) for 40 min. The speed and temperature were set as 300 rpm/min and 26 ± 2 °C, respectively. Then the extract was centrifuged at 10,000× *g* for 10 min. The residue was extracted three times in a row. The supernatants were combined and evaporated to near dryness with a rotary evaporator (RV8, IKA, Staufen, Germany) under vacuum at 60 °C, then redissolved with methanol to make the volume 50 mL.

### 2.3. Determination of α-Glucosidase Inhibitory Activity

The α-glucosidase inhibitory activity of the FSI samples was measured using the method described by Xiao et al. [13], with minor modifications. Each sample solution (0.2 mL) was mixed with 1 mL of α-glucosidase (0.2 U/mL) of pH = 6.8 phosphate buffer solution. After incubation at 37 °C for 15 min, 0.5 mL of 5 mmol/L PNPG in phosphate buffer solution was added. The reaction was terminated by the addition of 1 mL of 0.1 mol/L sodium Na_2_CO_3_ before incubation at 37 °C for 15 min. The absorbance was measured at 405 nm. The α-glucosidase inhibitory activity was calculated using the following Equation (1):(1)$$\alpha -Glucosidase\;inhibitory\;activity\left(\%\right)=\left(\frac{\Delta A_{control}-\Delta A_{sample}}{\Delta A_{control}}\right)\times 100$$

### 2.4. Omission Experiment

Three different solutions were prepared based on different polyphenol fractions (RU (3.40 mg/mL), QU (0.38 mg/mL), HY (0.34 mg/mL), QI (0.23 mg/mL) and KA (0.23 mg/mL): (1) Model solution: a model solution based on the composition of FSI extract. (2) Single polyphenol solution: a component solution containing only a single polyphenol, prepared based on the polyphenol content in FSI extract. (3) Recombinant solution: a reconstituted solution consisting of the missing individual polyphenol. The α-glucosidase inhibitory activities of the model, single polyphenol and recombinant solutions were determined and the contribution of the missing polyphenols to α-glucosidase activity inhibition was calculated [14].

### 2.5. Interaction Assay

Different combinations of polyphenols (0.02 mg/mL) were investigated according to the method of Li et al. [12]. Va denotes the α-glucosidase inhibition rate of polyphenol “a”, and “b, c, d, e” are named the same as “a”. V*^2^ was defined as the sum of the inhibition rates of two polyphenols, and the α-glucosidase inhibition rates of three (V*^3^), and four (V*^4^) were named in the same way as V*^2^. Vab is defined as the rate of α-glucosidase inhibition with the addition of polyphenols a and b, and the α-glucosidase inhibition rates of Vabc, and Vabcd were named in the same way as Vab.. Vmin represents the minimum value of α-glucosidase inhibition by a single polyphenol, while Vmax represents the maximum value. V*-Vab < −0.10 represents synergistic (SY). −0.1 < V*-Vab < 0.1 represents additive (AD). Sub-additive is represented by 0.1 < V*-Vab < Vmin (SU). Vmin < V*-Vab < Vmax for interference (IN). Vmax < V*-Vab for antagonism (AN).

### 2.6. α-Glucosidase Inhibition Type and Kinetic Analysis

In 5 mL of the reaction system, different concentrations of PNPG (0.5 mL) were mixed with different concentrations of polyphenols (0.1 mL) in a warm bath at 37 °C for 5 min, α-glucosidase solution (1 mL) was added and the absorbance was recorded at 405 nm in real time. Lineweaver–Burk curve method using the inverse of the PNPG concentration 1/(s) as the horizontal coordinate and the inverse of the enzymatic reaction rate 1/v as the vertical coordinate, and the type of inhibition was determined from the characteristics of the graph [15]. The inhibition type and kinetic constants were calculated by (2)–(4):(2)$$\frac{1}{v}=\frac{k_{m}}{v_{max}}\left(1+\frac{\left[I\right]}{K_{i}}\right)\frac{1}{\left[s\right]}+\frac{1}{V_{max}}$$
(3)$$Slope=\frac{k_{m}}{v_{max}}\left(1+\frac{\left[I\right]}{K_{i}}\right)$$
(4)$$Y_{intercept}=\frac{1}{V_{max}}\left(1+\frac{\left[I\right]}{K_{is}}\right)$$

### 2.7. Fluorescence Measurements

An equal volume of polyphenol solution (0.02 mg/mL) was added to 3 mL of α-glucosidase solution (0.05 mg/mL) in 9 separate additions of 10 μL each. The mixture was shaken and then placed in a warm bath at 37 °C for 15 min. The system was scanned for fluorescence at three temperatures (298 K, 303 K, 310 K) using an F-7000 fluorescence spectrometer (Hitachi, Tokyo, Japan) at an excitation wavelength of 280 nm and excitation and emission slits of 5 nm. The fluorescence was scanned in the range of 310–500 nm and the corresponding blank control fluorescence spectra were subtracted to calculate the fluorescence. The Stern–Volmer equation was used to calculate the fluorescence burst constants and determine the type of burst [16].

### 2.8. Thermodynamic Parameters

The binding reaction of the inhibitor and α-glucosidase is temperature dependent. Therefore, the spontaneity and binding force of their interactions can be studied by calculating thermodynamic parameters [13]. The values of the free energy change (Δ*G*), enthalpy change (Δ*H*) and entropy change (Δ*S*) can be calculated by Equations (5) and (6).
(5)$$\Delta G=-2.303RTlnK_{b}=\Delta H-T\Delta S$$
(6)$$ln\frac{K_{b2}}{K_{b1}}=\left(\frac{1}{T_{1}}-\frac{1}{T_{1}}\right)\Delta H\times \frac{1}{R}$$
where *R* = 8.314 J/(mol·K).

### 2.9. Circular Dichroism (CD) Analysis

α-glucosidase (0.3 mg/mL) with different polyphenols (1 mg/mL) was placed in a 400 μL cuvette and CD spectra (wavelength 190–260 nm) were collected using a Chirascan V100 spectrometer (Applied photophysics Ltd., Surrey, UK) with a control of 20 mM phosphate buffer (pH 6.8). The CD spectra were analysed using CDNN 2.1 software to obtain α-helix, β-sheet, β-turn and random coil [17].

### 2.10. Isothermal Titration Calorimetry (ITC) Analysis

The inhibitors RU, QU, HY, QI and KA were dissolved in 5% ethanol solution and loaded into VP-ITC (MicroCal; Malvern Panalytical, Malvern, UK) syringes. A solution of α-glucosidase (20 μM) dissolved in 5% ethanol solution was added to the ITC cell. The titration reaction temperature was 25 °C, the number of drops was set to 28, the stirring speed was set to 394 rpm and the reference power was set to 10.00 μcal/s. The titration interval was set to 180 s. The raw data peaks obtained were transformed into a relationship between the molar ratio and the enthalpy change per mole of injected solution [18].

### 2.11. Molecular Docking

Molecular docking was performed using AutoDock Vina1.1.2 [19]. α-glucosidase related parameters were set to: center_x = −9.293, center_y = −12.499, center_z = 11.342. The rest of the parameters were set by default. The 3D structure of the small molecule was downloaded in sdf format from the PubChem database according to its CAS number. The α-glucosidase (PDB ID: 3WY3) protein structure was downloaded from the PDB database; Pymol2.3.0 was used to remove protein crystalline water, primitive ligands, etc. The small molecule and protein structure were imported and saved into AutoDocktools (v1.5.6) for hydrogenation, charge calculation, charge assignment and to specify atom type. Docking results were analysed for interaction patterns using PyMOL 2.3.0 and LIGPLOT V 2.2.4.

### 2.12. Data Analysis

Duncan’s multivariate range test in SPSS software (Version 20.0, IBM, Armonk, New York, USA) was used for statistical analysis. The statistical significance of *p* < 0.05 was evaluated by one-way ANOVA. All determinations were repeated three times.

## 3. Results and Discussion

### 3.1. Key α-Glucosidase Inhibitors Were Identified by IC_50_ Value and Omission Test of Polyphenolic Compounds in FSI

According to previous studies, FSI contains the following polyphenols: rutin, quercetin, hyperoside, quercitrin, kaempferol, gentianoside, protocatechuic acid, isorhamnetin, kaempferol-3-O-rutinoside, narcissoside and chlorogenic acid [12]. To further compare the α-glucosidase inhibitory activity of these polyphenolic compounds, their IC_50_ values were calculated (Table 1). The positive control acarbose showed strong α-glucosidase inhibitory activity with an IC_50_ of 0.31 ± 0.01 mg/mL. Compared to acarbose, QU, HY and KA showed significant inhibition of α-glucosidase with IC_50_ values of 0.21 ± 0.01, 12.77 ± 0.07 and 0.55 ± 0.01 mg/mL, respectively. QU even showed a stronger inhibitory activity than the clinical drug acarbose. QI and RU showed relatively weak inhibition with IC_50_ values of 25.37± 0.13 and 57± 0.32 mg/mL, respectively. While QU and KA showed stronger inhibitory activity on α-glucosidase than others according to the literature [13], this may be due to the fact that the inhibitory activity of polyphenolic compounds against α-glucosidase is related to the hydroxyl substitution on the benzene ring.

The strength of the α-glucosidase inhibitory activity of RU, QU, HY, QI and KA were further evaluated by omission experiments. As shown in Table 1, there were significant differences in the α-glucosidase inhibition rates of the polyphenols in the FSI extract. QU exhibited the highest α-glucosidase inhibition rate of FSI extracts, followed by KA, HY, QI and RU. In addition, there was no significant difference in the inhibition rate between the sample solution (90.31%) and the model solution (90.55%) (Table 2), indicating that the model solution had mimicked the major polyphenols in the FSI extract better. The contribution of quercetin was significantly higher than that of other polyphenols. The α-glucosidase inhibition rate of the polyphenolic compounds without quercetin was 69.92%. However, in the presence of quercetin, the inhibition rate of the model solution was essentially unchanged (89.25%), further indicating that quercetin was the major inhibitor in the FSI extract. The contributions of α-glucosidase inhibitors in FSI were in the order of quercetin, kaempferol, chrysin, quercetin and rutin. Quercetin played a key role in the α-glucosidase inhibition rate in FSI extracts, which was consistent with the IC_50_ results. Oboh et al. [20] showed that quercetin significantly reduced α-glucosidase activity in mice. Other polyphenols in FSI extracts include KA, HY, QU and RU, which also have a similar structure to QU [21], but they have different inhibitory effects on α-glucosidase, a phenomenon related to the content of polyphenols in FSI extracts and also to the structure of the polyphenols. Based on the results of IC_50_ values and omission experiments, rutin, quercetin, chrysin, quercetin and kaempferol had strong inhibitory effects on α-glucosidase, so these five polyphenols were selected and their interactions were further investigated.

### 3.2. Interactions Analysis

The α-glucosidase inhibitory activities of RU, QU, HY, QI and KA were analysed (Table 3). Based on the results of the IC_50_ values, the concentrations of the different polyphenols were selected (close to the IC_50_ values) and ensured that the sum of their inhibition did not exceed 100% [11]. Among the different polyphenol combinations, the SY and AD effects were not observed, and only SU, IN and AN were observed. The combination of QU with KA produced a SU effect and the combination with RU, HY and QI produced an IN effect. This phenomenon suggested that the combination of QU with KA has minimal interference with the α-glucosidase inhibitory activity, whereas RU, HY and QI interfere with QU and affect the α-glucosidase inhibition rate [22]. When two α-glucosidase inhibitors (including QU and KA) are present, the AN effect is produced upon the addition of RU, HY and QI. When three, four or five polyphenols co-exist, all of these moieties exhibit either IN or AN. Yu et al. [14] found that two or more polyphenols together produced the SY effect. This may be due to the different types of inhibition of α-glucosidase by different polyphenols (Table 3). Quercetin readily interacts with amino acid residues Asn301, which are important catalytic sites for α-glucosidase [23]. KA inhibits α-glucosidase activity by preventing PNPG from entering the active site [22]. RU, HY and QI occupy the active site to prevent the enzyme from binding to the substrate [13]. Therefore, the relationship between inhibition mechanisms and polyphenol interactions needs to be further clarified.

All polyphenols were at a concentration of 0.02 mg/mL. IN represents the interference effect, SU represents the subadditive effect and AN represents the antagonism effect; RU represents rutin, QU represents quercetin, HY represents hyperoside, QI represents quercitrin, KA represents, kaempferol.

### 3.3. Inhibition Types and Constants

The α-glucosidase inhibition types of RU, QU, HY, QI and KA were analysed using inhibition kinetic assays and Lineweaver–Burk plots. For irreversible inhibition, the inhibitor binds to the enzyme through a covalent bond, whereas for reversible inhibition, the inhibitor binds to the enzyme through a non-covalent bond [24]. Three types of inhibition are obtained in Figure 1, including mixed inhibition, competitive inhibition and non-competitive inhibition [25]. As shown in Figure 1A–E and Table 1, the Km values increased and Vmax values decreased with increasing concentrations of RU, QU, HY, QI and KA, and all straight lines were fitted to intersect at the second limit, indicating that the inhibition of α-glucosidase by RU, QU, HY, QI and KA was a mixed type of inhibition [5]. The slopes were linear with polyphenol content (shown in Figure 1A–E), indicating that there is a single binding site for polyphenols to α-glucosidase, mainly by hydrophobic forces [12]. The binding values of polyphenols to α-glucosidase (Ki) or α-glucosidase-PNPG (Kis) can be obtained from Equations (3) and (4) (Table 1). Kis is higher than Ki for QU, QI and KA, suggesting that these polyphenols bind more strongly to α-glucosidase than to α-glucosidase-PNPG [26]. In addition, the RU (Ki = 8.91 mM) and HY (Ki = 3.91 mM) were shown to be non-competitive inhibitors. Taken together, RU, QU, HY, QI and KA can be considered as hybrid inhibitors of α-glucosidase.

### 3.4. Fluorescence Quenching

Fluorescence burst experiments can be used to determine the possible burst mechanism between the enzyme and the inhibitor. Fluorescence burst mechanisms are classified as static and dynamic bursts. The type of burst can be determined from the values of Kq and Ksv with temperature [27]. The energy transfer, binding sites, and binding constants of polyphenols with α-glucosidase were obtained by measuring the fluorescence emission spectra of α-glucosidase at different temperatures and different concentrations (Figure 2 and Table 4). The fluorescence quenching was enhanced with the addition of RU, QU, HY, QI and KA, indicating a significant α-glucosidase inhibitory effect of the inhibitors (Figure 2A,D,G,J,M). Thus, RU, QU, HY, QI and KA may be bursting agents of α-glucosidase fluorescence, altering the specific structural changes of α-glucosidase. We speculate that the fluorescent residues in α-glucosidase are buried inside the hydrophobic structure due to the induction of RU, QU, HY, QI and KA [5]. Furthermore, the addition of RU, QU, HY, QI and KA kept the characteristic fluorescence emission peaks intact.

Stern–Volmer curves were prepared for α-glucosidase inhibitors at 298, 303 and 310 K to explore possible burst mechanisms (Figure 2B,E,H,K,N). The Stern–Volmer curve showed a good linear correlation, indicating a single type of inhibition between the inhibitor and α-glucosidase [28]. Furthermore, the Ksv values increased from 298 K to 310 K, suggesting that the burst mechanism of RU, QU, HY, QI and KA may be dynamic. However, the Kq values were much higher than the maximum scattering collisional burst constants listed in Table 4 (2 × 10^12^ L mol^−1^s^−1^), suggesting that the burst mechanism was a complex formation or static burst [29]. The Stern–Volmer plots were well fitted, further suggesting that RU, QU, HY QI, and KA have static bursts with α-glucosidase. In addition, the binding sites for RU-α-glucosidase, QU-α-glucosidase, HY-α-glucosidase, QI-α-glucosidase and KA-α-glucosidase ranged from 1.57~2.35, indicating that only one or one class of binding sites was present. A higher K_b_ value indicates a stronger binding ability of the inhibitor with α-glucosidase, which means that the complex is more stable at higher temperatures [15]. The strongest binding capacity was for KA, followed by QU, QI, HY and RU.

### 3.5. Thermodynamic Parameters

The thermodynamic parameters of RU, QU, HY, QI and KA with α-glucosidase were calculated according to Equations (5) and (6) (Table 4). Electrostatic interactions, hydrogen bonding, hydrophobic forces and van der Waals forces are the four main non-covalent interactions between small molecules and biopolymers [30]. The positive values of Δ*H* and negative values of Δ*G* for the binding reactions of RU, QU, HY, QI and KA with α-glucosidase indicate that the binding process was spontaneous and that hydrophobic interactions were the main role in the binding process or that a conformational shift occurs [31]. In addition, the negative Δ*S* values for QU, QI and KA may be a result of weak hydrophobic interactions with α-glucosidase during binding (e.g., non-covalent van der Waals forces and hydrogen bonding) [32]. This is probably caused by the formation of hydrogen bonds, which reduced hydrophilicity and increased hydrophobicity [33]. This is consistent with the IC_50_ results. It remains to be investigated whether the inhibitor effect on the enzyme results in a conformational shift.

### 3.6. Circular Dichroism (CD) Analysis

CD spectroscopy is a powerful method to analyze the effect of inhibitors such as RU, QU, HY, QI and KA on the secondary structure of α-glucosidase and the conformational changes in the enzyme can be reflected in the corresponding CD spectra [17]. Figure 3A shows the CD spectra of RU, QU, HY, QI and KA at a concentration of 0.31 mg/mL. Based on the characteristics of the protein, α-glucosidase had two negative bands near both 208 nm and 220 nm. Due to the interaction of RU, QU, HY, QI and KA with α-glucosidase, the CD spectra showed a large variation from that of α-glucosidase, indicating that the binding process has a slight effect on the secondary structure of α-glucosidase. In order to quantify the secondary structure changes after α-glucosidase binding, the a-helix content of RU, QU, HY, QI and KA was estimated according to the two equations described in Reference [34]. Application of the above equations gave 63.6% of a-helix in free RU, QU, HY, QI and KA (Figure 3B), which is consistent with the previous study [35]. After adding 0.31 mg/mL of RU, QU, HY, QI and KA, the a-helix of RU, QU, HY, QI and KA decreased to 61.9%, 59%, 59.1%, 58% and 55.7%, respectively. The β-turn content decreased from 16.4% to 16.1%, 16.3%, 16.2%, 16.3%, and 16.3%, respectively, which may be due to the enzyme binding. This may be due to the tighter structure of the enzyme binding. This phenomenon suggests that RU, QU, HY, QI and KA caused a slight perturbation of the protein folding, although the slight decrease in a-helix content may not affect the physiological function of this carrier protein. The polypeptide chain may become soft in a specific way by accommodating RU, QU, HY, QI and KA molecules within the protein [29]. These results suggest that RU, QU, HY, QI and KA induce secondary structural slight changes in α-glucosidase, and have an insignificant effect on α-glucosidase activity [36].

### 3.7. Isothermal Titration Calorimetry (ITC) Analysis

ITC is an effective method for calculating thermal changes and characterizing the mechanism of interaction between two molecules [37]. The binding affinity, and entropic and enthalpy changes of the five inhibitors with α-glucosidase were studied using the ITC method. Figure 4 shows the curves of the heat of binding of the five inhibitors per titration of α-glucosidase alone and the heat of integration released per titration as a function of the molar ratio of the inhibitor to α-glucosidase. The software in the ITC instrument allows the calculation of the parameters Δ*H*, Δ*S*, binding stoichiometry (n) and equilibrium dissociation constant (KD) (Table 5). Figure 4A shows that the binding process of RU to α-glucosidase is heat-absorbing, as the graph has a positive peak [38]. Based on the calculation of the values of Δ*H* and Δ*S* (Table 5), it was clear that Δ*G* was less than 0 and Δ*S* was greater than 0. RU binding to α-glucosidase is mainly entropy-driven and spontaneous, and Δ*H* and Δ*S* were both greater than 0. Hydrophobic interactions may play a considerable role in the binding process [18]. In contrast, the plot of QU with α-glucosidase shows a negative peak, and the binding process is exothermic (Figure 4B). Δ*G* was less than 0 and Δ*S* was greater than 0. The binding of QU with α-glucosidase is a mainly entropy-driven spontaneous interaction, with Δ*S* greater than 0 and Δ*S* less than 0. The interaction occurs mainly through hydrophobic forces and the formation of a few hydrogen bonds [16]. The binding process of HY to α-glucosidase was heat-absorbing similar to that of RU (Figure 4C). Δ*G* was less than 0 and Δ*S* was greater than 0. Δ*H* and Δ*S* were both greater than 0, and mainly entropy-driven spontaneous interactions; hydrophobic interactions may play a major role in the binding. The binding of QI to α-glucosidase was exothermic (Figure 4D). When Δ*G* was less than 0, the reaction proceeded spontaneously; when Δ*S* was greater than 0 and Δ*H* was less than 0, the interaction occurred mainly through hydrophobic forces and the formation of a few hydrogen bonds. The binding of KA to α-glucosidase was similar to QU and QI and was exothermic (Figure 4E); Δ*S* was greater than 0, while Δ*H* was less than 0. The interaction occurred mainly through hydrophobic forces as well as the formation of a few hydrogen bonds [19]. This result was consistent with the thermodynamic findings. Further studies on which moiety of the inhibitor and enzyme react are pending.

### 3.8. Molecular Docking Analysis

Molecular docking is an effective method to analyse the interaction of inhibitors with enzyme moieties [39]. In this experiment, Autodock was employed to investigate the binding modes of RU, QU, HY, QI and KA to α-glucosidase [40]. α-Glucosidase binds to RU, QU, HY, QI and KA at energies of −8.0, −8.1, −7.9, −8.3 and −7.50 kcal/mol, respectively. As shown in Figure 5A, RU mainly forms hydrogen bonds with α-glucosidase amino acid residues Asn301 with hydrogen bond lengths of 3.05 Å, 2.89 Å, 3.31 Å; and with Pro230, Leu227, Thr226, Ala229, Glu231, Ala232, Glu396, Lys398, Phe397, Val335, Met302 and Arg340. QU mainly forms hydrogen bonds with the α-glucosidase amino acid residues Asn301, with a hydrogen bond length of 3.70 Å; with Met302, Val335, Glu396, Glu396, Glu196, Asn301, Asn301, Glu396, Phe397, Pro2230, Gly228, Leu227, Val334, Phe297 and Leu300 with hydrophobic interactions (Figure 5B). HY forms hydrogen bonds with the α-glucosidase amino acid residues Ala232 and Asn301 with hydrogen bond lengths of 2.93 Å, 3.80 Å and 3.15 Å, respectively; and Glu231, Pro230, Leu300, Met302, Gly228, Ala229, Phe297, Val334, Val335, Arg340, Gly339, Lys398 and Phe397 are hydrophobic (Figure 5C). QI mainly forms hydrogen bonds with α-glucosidase amino acid residues Val335 and Asn301, with hydrogen bond lengths of 3.69 Å, 3.20 Å and 3.46 Å, respectively; with Glu231, Lys225, Thr226, Pro230, Ala229, Arg340, Val334, Asp333, Phe397 and Glu396 with hydrophobic interactions (Figure 5D). KA mainly interacts with the α-glucosidase amino acid residues Val380, Asp401 and GLy402 with hydrogen bond lengths of 3.18 Å, 2.84 Å and 2.99 Å respectively; and hydrophobic interactions with Asp379, Thr339, Val335, Glu377, Gly399, Arg400 and Lys398 (Figure 5E). The molecular docking results showed that RU, QU, HY, QI and KA all entered the cavity of α-glucosidase, mainly affected the hydrophobic forces and formed a few hydrogen bonds [41]. This was consistent with the previous thermodynamic analysis and ITC results.

## 4. Conclusions

In this study, RU, QU, HY, QI and KA were identified as the primary α-glucosidase inhibitors and mixed inhibitors in FSI. Quercetin has the strongest α-glucosidase inhibition ability (IC_50_ = 0.212 mg/mL), followed by kaempferol, hypericin, quercetin and kaempferol. Meanwhile,, the combination of QU with KA produced a SU effect and the combination with RU, HY and QI produced an IN effect. Moreover, RU, QU, HY, QI and KA had a significant fluorescence burst effect on α-glucosidase and the interaction mechanism was a static process. However, circular dichroism spectroscopy indicated that the inhibitor induced slight secondary structural changes in α-glucosidase, and had an insignificant effect on α-glucosidase activity. Furthermore, ITC analysis and molecular docking results indicated that the binding of the inhibitor to the enzyme was entirely spontaneous, with the polyphenols interacting with α-glucosidase through hydrogen bonding and hydrophobic interactions, with hydrophobic interactions being the main driving force. The results of this study provide new insights and valuable information for the study of the inhibitory mechanism of α-glucosidase by FSI polyphenols. Further in vivo experiments are needed in the future to demonstrate their hypoglycemic activity and application mechanism.

## Figures and Tables

**Figure 1 foods-12-00715-f001:**
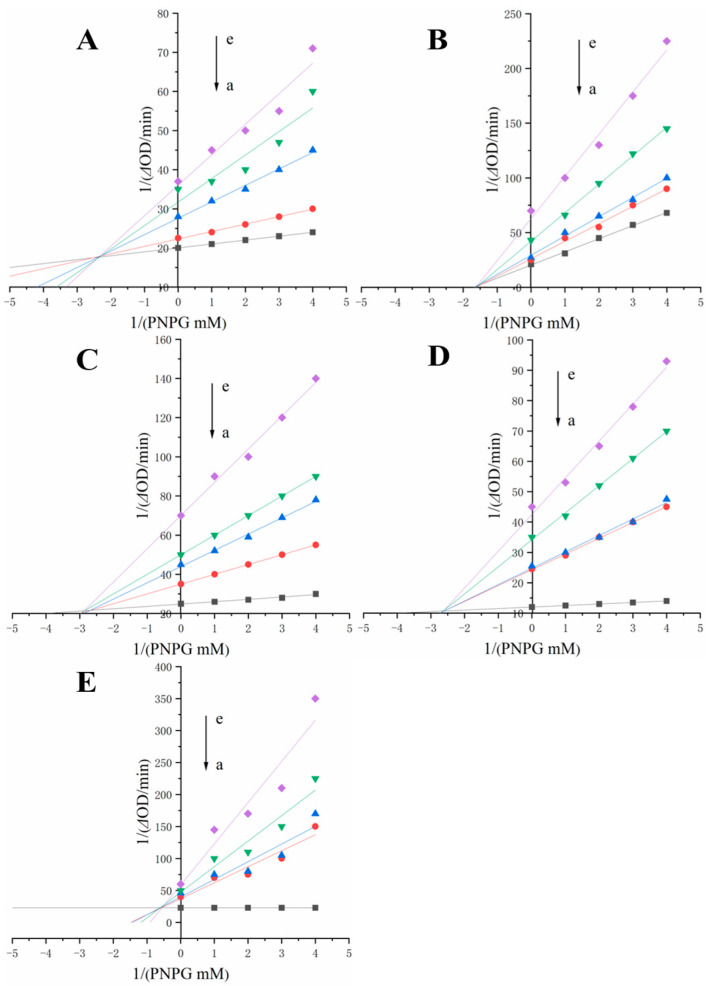
Lineweaver–Burk plot of different inhibitors against α-glucosidase. (**A**), RU, a–e concentrations were 0, 0.33, 0.66, 1.3, 2.6 mM; (**B**), QU, a–e concentrations were 0, 0.33, 0.66, 1.3, 2.6 mM, respectively; (**C**), HY, a–e concentrations were 0, 0.33, 0.66, 1.3, 2.6 mM; (**D**), QI, a–e concentrations were 0, 0.33, 0.66, 1.3, 2.6 mM; (**E**), KA, a–e concentrations were 0, 0.33, 0.66, 1.3, 2.6 mM.RU represents rutin, QU represents quercetin, HY represents hyperoside, QI represents quercitrin, KA represents, kaempferol.

**Figure 2 foods-12-00715-f002:**
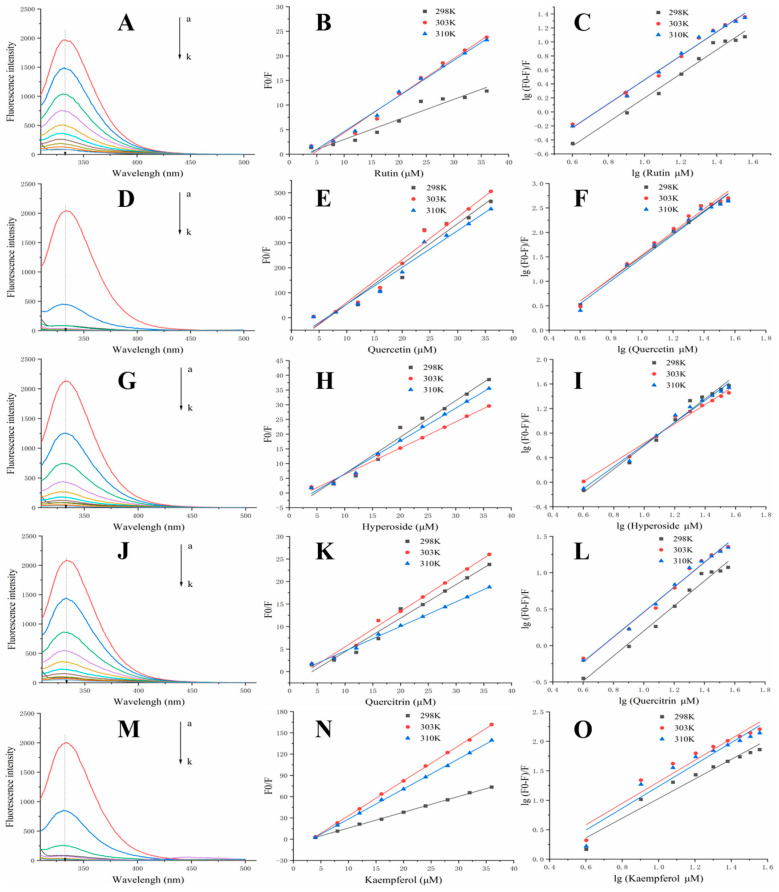
Fluorescence spectra and values of α-glucosidase in the addition of inhibitors under different concentrations ((**A**–**C**) represent RU; (**D**–**F**) represent QU; (**G**–**I**) represent HY; (**J**–**L**) represent QI; (**M**–**O**) represent KA)), a–k concentrations were 10 μL of polyphenol solution (0.02 mg/mL) were added in 11 portions. RU represents rutin, QU represents quercetin, HY represents hyperoside, QI represents quercitrin, KA represents, kaempferol.

**Figure 3 foods-12-00715-f003:**
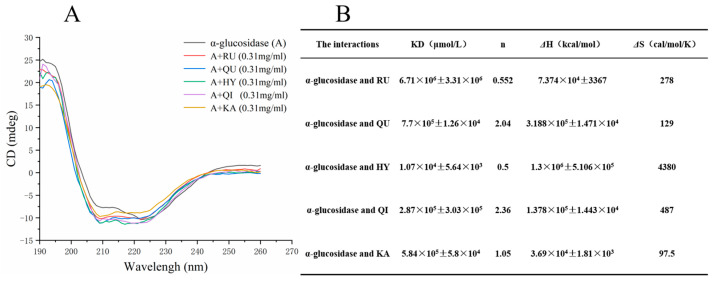
CD spectra of α-glucosidase with RU, QU, HY, QI, KA (**A**). The contents of secondary structures of α-glucosidase in absence or presence of RU, QU, HY, QI, KA at pH = 6.9 T = 298 K (**B**). RU represents rutin, QU represents quercetin, HY represents hyperoside, QI represents quercitrin, KA represents, kaempferol.

**Figure 4 foods-12-00715-f004:**
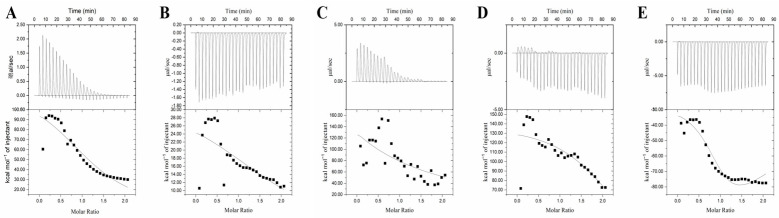
Isothermal titration calorimetry profiles of the RU (**A**), QU (**B**), HY (**C**), QI (**D**), KA (**E**) and α-glucosidase. RU represents rutin, QU represents quercetin, HY represents hyperoside, QI represents quercitrin, KA represents, kaempferol.

**Figure 5 foods-12-00715-f005:**
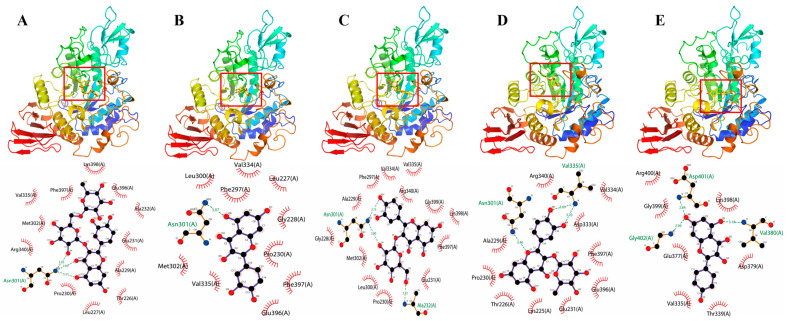
Three-dimensional and two-dimensional docking modes of RU (**A**), QU (**B**), HY (**C**), QI (**D**) and KA (**E**) with α-glucosidase by molecular docking. RU represents rutin, QU represents quercetin, HY represents hyperoside, QI represents quercitrin, KA represents, kaempferol.

**Table 1 foods-12-00715-t001:** The values of IC_50_ and inhibitory constants of polyphenols in *Flos sophorae immaturus* extract on α-glucosidase inhibitory activity. Different letters in the table indicate that the IC_50_ values are significantly different (*p* < 0.05).

Compounds	IC_50_ (mg/mL)	Ki (mM)	Kis (mM)
Protocatechuic acid	369.00 ± 0.11 ^h^	17.71	-
Chlorogenic acid	-	-	-
Rutin	57.00 ± 0.32 ^f^	8.91	-
Hyperoside	12.77 ± 0.07 ^d^	3.79	-
Kaempferol-3-O-rutinoside	-	-	-
Narcissoside	-	-	-
Quercitrin	25.37 ± 0.13 ^e^	5.31	6.72
Quercetin	0.21 ± 0.01 ^a^	0.22	2.51
Kaempferol	0.55 ± 0.01 ^c^	0.67	5.56
Isorhamnetin	255.00 ± 0.11 ^g^	15.12	-
Acarbose (positive control)	0.31 ± 0.01 ^b^	0.57	-

**Table 2 foods-12-00715-t002:** α-Glucosidase inhibition rates of omitted compounds and recombinant solutions, and contribution rates of omitted compounds to the α-glucosidase inhibition rate of the model solution.

Omitted Compounds	α-Glucosidase Inhibition Rate of Omitted Compounds (%)	α-Glucosidase Inhibition Rate of Recombinant Solutions Omitting Different Compounds (%)	Contribution Rates of Omitted Compounds to the α-Glucosidase Inhibition Rate of Model Solution (%)
RU	69.00 ± 0.11	85.81 ± 0.15	4.70 ± 0.13
QU	89.00 ± 0.25	69.92 ± 2.90	20.59 ± 0.26
HY	75.00 ± 0.32	81.90 ± 0.59	8.61 ± 0.16
QI	77.00 ± 0.20	83.79 ± 0.25	6.72 ± 0.08
KA	81.00 ± 0.12	75.89 ± 0.18	14.62 ± 0.17
None (Model solution)		90.55 ± 0.11	
None (Sample solution)		90.31 ± 0.21	

RU represents rutin, QU represents quercetin, HY represents hyperoside, QI represents quercitrin, KA represents, kaempferol.

**Table 3 foods-12-00715-t003:** Interaction effects of polyphenols in FSI extract on the α-glucosidase inhibition rate.

Group	Va + Vb	Value		Interaction
		V*^2^	Vab	V*^2^-Vab
1	RU + QU	0.6556	0.3567	0.2989 IN
2	RU + HY	0.6321	0.3356	0.2965 IN
3	RU + QI	0.5527	0.2986	0.2541 IN
4	RU + KA	0.5789	0.3025	0.2764 IN
5	QU + HY	0.6851	0.3799	0.3052 IN
6	QU + QI	0.6677	0.3657	0.3020 IN
7	QU + KA	0.6956	0.3966	0.2990 SU
8	HY + QI	0.6425	0.3485	0.2940 IN
9	HY + KA	0.6521	0.3599	0.2922 IN
10	QI + KA	0.6399	0.3567	0.2832 IN
Group	Va + Vb + Vc	V*^3^	Vabc	V*^3^-Vabc
11	RU + QU + HY	0.7225	0.4263	0.2962 AN
12	RU + QU + QI	0.7100	0.4130	0.2970 AN
13	RU + QU + KA	0.7525	0.4536	0.2989 AN
14	RU + HY + QI	0.6977	0.3935	0.3042 IN
15	RU + HY + KA	0.7122	0.4001	0.3121 IN
16	RU + QI + KA	0.7055	0.3966	0.3089 IN
17	QU + HY + QI	0.7369	0.4358	0.3011 IN
18	QU + HY + KA	0.7768	0.4963	0.2805 IN
19	QU + QI + KA	0.7588	0.4832	0.2756 IN
20	HY + QI + KA	0.7322	0.4255	0.3067 IN
Group	Va + Vb + Vc + Vd	V*^4^	Vabcd	V*^4^-Vabcd
21	RU + QU + HY + QI	0.8898	0.5362	0.3536 AN
22	RU + QU + HY + KA	0.9012	0.5203	0.3809 AN
23	RU + QU + QI + KA	0.8922	0.5136	0.3786 IN
24	RU + HY + QI + KA	0.8725	0.5001	0.3724 IN
25	QU + HY + QI + KA	0.9556	0.5656	0.3900 IN

**Table 4 foods-12-00715-t004:** Effect of RU, QU, HY, QI, KA on the binding constants, binding sites and thermodynamic parameters at 298, 303 and 310 K.

Compound	T (K)	K_SV_ (×10^4^ L mol^−1^)	Kq (×10^12^ L mol^−1^ s^−1^)	Kb (×10^5^ L mol^−1^)	n	R^2^	Δ*H* (KJ mol^−1^)	Δ*G* (KJ mol^−1^)	Δ*S* (J mol^−1^ K^−1^)
RU	298	4.50	4.50	0.52	1.72	0.99	28.48	−2.39	31.72
	303	9.37	9.37	0.26	1.71	0.99		−1.32	27.70
	310	9.34	9.34	0.25	1.71	0.99		−1.33	27.08
QU	298	10.71	10.71	1.46	2.30	0.99	−5.34	−4.46	8.75
	303	7.52	7.52	1.44	2.35	0.99		−4.64	8.01
	310	5.55	5.55	1.42	2.34	0.99		−4.94	6.87
HY	298	12.43	12.43	0.77	1.91	0.99	8.06	−7.49	1.41
	303	8.96	8.96	0.79	1.57	0.99		−5.41	5.48
	310	11.17	11.17	0.71	1.80	0.99		−7.04	0.09
QI	298	7.45	7.45	0.62	1.72	0.99	−40.30	−61.22	181.73
	303	7.87	7.87	0.68	1.71	0.99		−59.72	176.69
	310	5.45	5.45	0.69	1.71	0.99		−59.69	176.58
KA	298	22.23	22.23	1.22	1.67	0.99	−18.49	−69.32	208.92
	303	49.33	49.33	1.26	1.83	0.99		−68.62	203.15
	310	42.46	42.46	1.25	1.86	0.99		−69.23	200.53

RU represents rutin, QU represents quercetin, HY represents hyperoside, QI represents quercitrin, KA represents, kaempferol.

**Table 5 foods-12-00715-t005:** Parameters of RU, QU, HY, QI, KA binding to α-glucosidase as determined by isothermal titration calorimetry.

The Interactions	KD (μmol/L)	n	Δ*H* (kcal/mol)	Δ*S* (cal/mol/K)
α-glucosidase and RU	1.24 × 10^5^ ± 6.5 × 10^4^	1.34	1.212 × 10^5^ ± 2.145 × 10^4^	430
α-glucosidase and QU	7.7 × 10^5^ ± 1.26 × 10^4^	2.04	−3.188 × 10^5^ ± 1.471 × 10^4^	129
α-glucosidase and HY	1.07 × 10^4^ ± 5.64 × 10^3^	0.5	1.3 × 10^6^ ± 5.106 × 10^5^	4380
α-glucosidase and QI	2.87 × 10^5^ ± 3.03 × 10^4^	2.36	−1.378 × 10^5^ ± 1.443 × 10^4^	487
α-glucosidase and KA	5.84 × 10^5^ ± 5.8 × 10^4^	1.05	−3.69 × 10^4^ ± 1.81 × 10^3^	97.5

RU represents rutin, QU represents quercetin, HY represents hyperoside, QI represents quercitrin, KA represents, kaempferol.

## Data Availability

The data presented in this study are available on request from the corresponding author.

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
