# Peer review of "In Vitro Inhibitory Effects of Polyphenols from Flos sophorae immaturus on α-Glucosidase: Action Mechanism, Isothermal Titration Calorimetry and Molecular Docking Analysis"

_foods, 2023, doi:10.3390/foods12040715_

Round 1
Reviewer 1 Report
Manuscript titled “In vitro inhibitory effects polyphenols of from Flos Sophorae Immaturus on α-glucosidase: Action mechanism, isothermal titration calorimetry and molecular docking analysis" reports various in vitro and in silico analyses aimed at determining the glucosidase-inhibition potential of Flos Sophorae Immaturus. The authors used complementary methods to determine which compounds and/or their combinations were responsible for the observed effects, their behavior in isolation or in combination, and docking analyses to further study the observed phenomenon. There are some comments for the authors:
1. In the title, change “polyphenols of” to “of polyphenols”.
2. In line 34, rutin content is highlighted as “reaching 20%”. Do you mean rutin is 20% of its phenolics, or 20% of the tissue is rutin or something else? Please clarify your intended meaning.
3. Please define “PNPG” in line 61.
4. Is the glucosidase used in the present work of porcine origin (line 61)? Please specify.
5. Please specify the phenolic concentration used in sections 2.3-2.6. Some sections mention the volumes added, but not concentrations.
6. What do you mean by “brassin” in line 160?
7. “Quercetin played a key role in the inhibition of α-glucosidase in low concentration FSI extracts, which was consistent with the IC50 results”. Please clarify your intended meaning of this sentence, if quercetin was shown to be the major contributor to the observed effect, why would it not play a role when administered at low concentrations? Or perhaps my interpretation of this phrase is inaccurate. Also, to what low concentration FSI extracts are you referring? It is not immediately clear
8. In tables 1 and 2, please homogenize the number of decimal digits for a data point and its error. For example, IC50 for protocatechuic acid is 369 +/- 0.11; homogenize to zero, one or two decimal digits for both numbers. Similar comment for table 4, where data are shown with different number of decimal digits.
9. Most concentrations shown in table 3 are 0.02 mg/mL, please consider mentioning this on the table’s footer instead of repeating it. Only values that differ could be mentioned, in order to make the table easier to read.
10. Line 329 mentions “Supplementary 4”, do you mean figure 4?
11. Section 3.8 repeats “(A)” various times when mentioning the amino acids involved. What does this indicate? It is not immediately clear.
12. Please consider having your manuscript revised to amend various typos and writing mistakes. This should ideally be done by a native English-speaking colleague or professional service.
Author Response
Reviewer 1:
Manuscript titled “In vitro inhibitory effects polyphenols of from Flos Sophorae Immaturus on α-glucosidase: Action mechanism, isothermal titration calorimetry and molecular docking analysis" reports various in vitro and in silico analyses aimed at determining the glucosidase-inhibition potential of Flos Sophorae Immaturus. The authors used complementary methods to determine which compounds and/or their combinations were responsible for the observed effects, their behavior in isolation or in combination, and docking analyses to further study the observed phenomenon. There are some comments for the authors.
Response: We very much appreciate the careful reading of our manuscript and the valuable suggestions of the reviewers. We have carefully considered the comments and have revised the manuscript accordingly. The answer to each question is listed below.
- In the title, change “polyphenols of” to “of polyphenols”.
Response:According to the reviewer’s suggestion, the “polyphenols of” has been revised to “of polyphenols” in line 2.
- In line 34, rutin content is highlighted as “reaching 20%”. Do you mean rutin is 20% of its phenolics, or 20% of the tissue is rutin or something else? Please clarify your intended meaning.
Response:Thank you so much for your valuable comments, the “reaching 20%” has been revised to “with rutin content up to 20% of the total mass of FSI” in line 49-50.
- Please define “PNPG” in line 61.
Response:According to the reviewer’s suggestion, the “PNPG” has been revised to “4-nitrophenyl β-D-glucopyranoside (PNPG)” in line 91.
- Is the glucosidase used in the present work of porcine origin (line 61)? Please specify.
Response:Thank you so much for your valuable comments, the “α-glucosidase” has been revised to “α-glucosidase (Yeast)” in line 91.
- Please specify the phenolic concentration used in sections 2.3-2.6. Some sections mention the volumes added, but not concentrations.
Response:Thank you so much for your valuable comments, we have added the phenolic concentration of sections 2.3-2.6 in line 116-118, 127, 294-297, 148.
- What do you mean by “brassin” in line 160?
Response:Thank you for the valuable suggestions, we have deleted this mistake in line 215.
- “Quercetin played a key role in the inhibition of α-glucosidase in low concentration FSI extracts, which was consistent with the IC50 results”. Please clarify your intended meaning of this sentence, if quercetin was shown to be the major contributor to the observed effect, why would it not play a role when administered at low concentrations? Or perhaps my interpretation of this phrase is inaccurate. Also, to what low concentration FSI extracts are you referring? It is not immediately clear.
Response:Thank you so much for your valuable comments, we have revised to “Quercetin played a key role in the α-glucosidase inhibition rate in FSI extracts, which was consistent with the IC50 results”. in line 230-232.
- In tables 1 and 2, please homogenize the number of decimal digits for a data point and its error. For example, IC50 for protocatechuic acid is 369 +/- 0.11; homogenize to zero, one or two decimal digits for both numbers. Similar comment for table 4, where data are shown with different number of decimal digits.
Response:Thank you so much for your valuable suggestion. we have homogenized the number of decimal digits for a data point and its error in tables 1, 2 and 4.
- Most concentrations shown in table 3 are 0.02 mg/mL, please consider mentioning this on the table’s footer instead of repeating it. Only values that differ could be mentioned, in order to make the table easier to read.
Response:According to the reviewer’s suggestion, we have deleted the “0.02 mg/mL’’ in table 3 and added to table’s footer in line 269.
- Line 329 mentions “Supplementary 4”, do you mean figure 4?
Response:Thank you so much for your valuable comments, the “Supplementary 4” has been revised to “Table 5” in line 390.
- Section 3.8 repeats “(A)” various times when mentioning the amino acids involved. What does this indicate? It is not immediately clear.
Response:Thank you so much for your valuable comments, we have deleted the “(A)” in line 427-445.
- Please consider having your manuscript revised to amend various typos and writing mistakes. This should ideally be done by a native English-speaking colleague or professional service.
Response:According to the reviewer’s suggestion, we have corrected the various typos and writing mistakes in line 6, 8, 10, 45-47, 49-50, 51-58, 73, 78-82, 104, 136-138, 148-149, 155, 166, 168, 201-205, 207-209, 211, 213, 215-217, 219-220, 223, 225-226, 228-232, 237-239, 248, 253-254, 256-257, 259-260, 274, 287, 304-308, 317, 325-326, 341-342, 345-347, 357-358, 360, 365-366, 368, 370, 374, 383-384, 386, 388, 391, 393-395, 400, 402, 404-405, 410-411, 423-425, 446-448, 455-458,462, 467.
Reviewer 2 Report
Comments to the Author
Title: In vitro inhibitory effects polyphenols of from Flos Sophorae Immaturus on α-glucosidase: Action mechanism, isothermal titration calorimetry and molecular docking analysis.
Title
Please correct the title “…effects polyphenols of from...” to “…effects of polyphenols from...”
Abstract
Please rewrite well the abstract because it doesn’t give a clear insight about the research work. Moreover, it is highly recommended to improve the English style of the overall manuscript.
Introduction
Please try to extend the introduction because it is very concise, providing previous works conducted on the alpha-glucosidase inhibition with other polyphenols and their references.
Materials and methods
Please describe well the plant extract preparation.
2.2 Determination of α-glucosidase inhibitory activity
Can the author explain why they didn’t monitor the pH of the test with the employment of the buffer.
2.3 Omission experiment
The author has described all employed reagents in the experiments, indicating that the plant material of Flos Sophorae Immaturus was purchased and more detailly explained on the Line 45 that this plant material contains different polyphenolic compounds according to the previous conducted study. How come that the author has led an omission experiment with single polyphenolic compound? Did the author used standard single polyphenols? If yes, it should be indicated first in the purchased reagents, and furtherly described in the experiment. Or did the author employed the preparative HPLC to separate each phenolic compound from the extract in order to exclude each single compound from the extract?
3. Results and discussion
Line 156, quercitrin and rutin don’t seem to present a moderate inhibition against alpha-glucosidase as indicated, because they less effective than the positive control with almost 84 and 195 folds, respectively. Without applying any statistical analyses for the obtained results would open a big debate on their effectiveness.
Author Response
Reviewer 2:
Comments to the Author:
Title: In vitro inhibitory effects polyphenols of from Flos Sophorae Immaturus on α-glucosidase: Action mechanism, isothermal titration calorimetry and molecular docking analysis.
Response: We very much appreciate the careful reading of our manuscript and the valuable suggestions of the reviewers. We have carefully considered the comments and have revised the manuscript accordingly. The answer to each question is listed below.
- Please correct the title “…effects polyphenols of from...” to “…effects of polyphenols from...”.
Response:Thank you so much for your valuable comments, the “effects polyphenols of from” has been revised to “effects of polyphenols from” in line 2.
- Please rewrite well the abstract because it doesn’t give a clear insight about the research work. Moreover, it is highly recommended to improve the English style of the overall manuscript.
Response:Thank you so much for your valuable comments. We have improved the abstract in line 14-29 and improved the English style of the overall manuscript in line 6, 8, 10, 45-47, 49-50, 51-58, 73, 78-82, 104, 136-138, 148-149, 155, 166, 168, 201-205, 207-209, 211, 213, 215-217, 219-220, 223, 225-226, 228-232, 237-239, 248, 253-254, 256-257, 259-260, 274, 287, 304-308, 317, 325-326, 341-342, 345-347, 357-358, 360, 365-366, 368, 370, 374, 383-384, 386, 388, 391, 393-395, 400, 402, 404-405, 410-411, 423-425, 446-448, 455-458,462, 467.
- Please try to extend the introduction because it is very concise, providing previous works conducted on the alpha-glucosidase inhibition with other polyphenols and their references.
Response:According to the reviewer’s suggestion, we have extended the introduction about of alpha-glucosidase inhibition with other polyphenols in line 58-67.
- Please describe well the plant extract preparation.
Response:Thank you so much for your valuable comments, we have added the plant extract preparation “Dried FSI (1.25 g) was extracted with 40 mL 70% (v/v) ethanol on a magnetic stirrer (C-MAG HS 7, IKA, Staufen, Germany) for 40 min. The speed and temperature were set as 300 rpm/min and 26 ± 2 °C, respectively. Then the extract was centrifuged at 10,000 g for 10 min. The residue was extracted three times in a row. The supernatants were combined and evaporated to near dryness with a rotary evaporator (RV8, IKA, Staufen, Germany) under vacuum at 60 °C, then re-dissolved with methanol to make the volume to 50 mL.”in line 94-101.
- 2.2 Determination of α-glucosidase inhibitory activity: Can the author explain why they didn’t monitor the pH of the test with the employment of the buffer.
Response:Thank you so much for your valuable comments, we have revised the determination of α-glucosidase inhibitory activity to “Each Sample solution (0.2mL) was mixed with 1mL of α-glucosidase (0.2 U/mL) of pH = 6.8 phosphate buffer solution. After incubation at 37°C for 15 min, 0.5 mL of 5 mmol/L PNPG in phosphate buffer solution was added. The reaction was terminated by the addition of 1 mL of 0.1 mol/L sodium Na2CO3 before incubation at 37°C for 15 min. The absorbance was measured at 405 nm.” in line 104-109.
- Omission experiment: The author has described all employed reagents in the experiments, indicating that the plant material of Flos Sophorae Immaturus was purchased and more detailly explained on the Line 45 that this plant material contains different polyphenolic compounds according to the previous conducted study. How come that the author has led an omission experiment with single polyphenolic compound? Did the author used standard single polyphenols? If yes, it should be indicated first in the purchased reagents, and furtherly described in the experiment. Or did the author employed the preparative HPLC to separate each phenolic compound from the extract in order to exclude each single compound from the extract?
Response:Thank you so much for your valuable comments, the omission experiment of polyphenols inhibiting α-glucosidase inhibitory activity was performed to determine the vital α-glucosidase inhibitory substances in FSI extract. And we have added the single polyphenols “rutin, quercetin, hyperoside, quercitrin, kaempferol, gentianoside, protocatechuic acid, isorhamnetin, kaempferol-3-O-rutinoside, narcissoside, and chlorogenic acid” in line 88-91.
- Line 156, quercitrin and rutin don’t seem to present a moderate inhibition against alpha-glucosidase as indicated, because they less effective than the positive control with almost 84 and 195 folds, respectively. Without applying any statistical analyses for the obtained results would open a big debate on their effectiveness.
Response:According to the reviewer’s suggestion, we have added the statistical analyses in table 1 and the “QI and RU showed moderate inhibition” has been revised to “QI and RU showed relatively weak inhibition” in line 211.
Reviewer 3 Report
This paper compares the inhibitory effects of some Flos Sophorae Immaturus polyphenols on α-glucosidase (more particularly as flavonoids, aglycone or in the form of heterosides) and helps to understand their inhibitory mechanism using binding assay, omission method, interaction, Fluorescence burst effect on α-glucosidase, circular dichroism spectroscopy to help understand α-glucosidase conformation changes and the interaction modes using isothermal titration calorimetry as well as in silico assay,
The paper is very good and well written, I could not find any mistakes, well done,
some minor points :
In the abstract
- line 18 Quercetin was strongest inhibition and contribution
- line 20, dot instead comma after effect
in material and method :
- line 67-70 : rephrase the paragraph : Add 0.5 ml of 5 mmol/L PNPG solution and react
in a water bath at 37°C for 10 min, then add 1 ml of 0.1 mol/L sodium Na2CO3 solution to
stop the reaction. Measure the absorbance value at 405nm
0.5 ml of 5 mmol/L PNPG solution is added…...
correct :
- The inhibition rate of α- glucosidase activity inhibition was determined for these three solutions
- rephrase the paragraph 135-140
- on table 1 add a note on acarbose that it is a positive control compared to polyphenols…
- table 3 to reorganize, as long as to decrease the font
- add reference at the beginning of the line 303
- paragraph 3.7, replace ITC by isothermal titration calorimetry
- rephrase the protocol of docking experiment :
Download the α- glucosidase (PDB ID: 3WY3) protein structure from the PDB database; use Pymol2.3.0 to remove protein crystalline water, primitive ligands, etc. Import the small molecule and protein structure into AutoDocktools (v1.5.6) for hydrogenation, charge calculation, charge assignment, specify atom type and save.
The α- glucosidase (PDB ID: 3WY3) structure was downloaded from the PDB……etc
- line 153 : Compared
- line 195 : rephrase « no SY and AD effects were found »
- line 196 : The combination of QU with
- correct unit weighting of binding constants in table 4
lines 344-346, rephrase and correct punctuation
Author Response
Reviewer 3:
This paper compares the inhibitory effects of some Flos Sophorae Immaturus polyphenols on α-glucosidase (more particularly as flavonoids, aglycone or in the form of heterosides) and helps to understand their inhibitory mechanism using binding assay, omission method, interaction, Fluorescence burst effect on α-glucosidase, circular dichroism spectroscopy to help understand α-glucosidase conformation changes and the interaction modes using isothermal titration calorimetry as well as in silico assay.
The paper is very good and well written, I could not find any mistakes, well done.
Response: We very much appreciate the careful reading of our manuscript and the valuable suggestions of the reviewers. We have carefully considered the comments and have revised the manuscript accordingly. The answer to each question is listed below.
- line 18, Quercetin was strongest inhibition and contribution.
Response: Thank you so much for your valuable comments, the “Quercetin was strongest inhibition and contribution” has been revised to “Quercetin plays a considerable a-glucosidase inhibition ability in FSI.” in line 20-21.
- line 20, dot instead comma after effect.
Response: Thank you for the valuable suggestions, the “comma” has been revised to “dot” in line 21.
- line 67-70: rephrase the paragraph: Add 0.5 mL of 5 mmol/L PNPG solution and react in a water bath at 37°C for 10 min, then add 1 mL of 0.1 mol/L sodium Na2CO3 solution to stop the reaction. Measure the absorbance value at 405 nm.
Response: Thank you so much for your valuable comments, we have revised to “0.5 mL of 5 mmol/L PNPG in phosphate buffer solution was added. The reaction was terminated by the addition of 1 mL of 0.1 mol/L sodium Na2CO3 before incubation at 37 °C for 15 min. The absorbance was measured at 405 nm.” in line 106-109.
- Correct: The inhibition rate of α- glucosidase activity inhibition was determined for these three solutions.
Response: According to the reviewer’s suggestion. we have revised to the “The α-glucosidase inhibitory activities of the model, single polyphenol, and recombi-nant solutions were determined” in line 121-123.
- Rephrase the paragraph 135-140.
Response: Thank you for the valuable suggestions. We have revised to “T The α-glucosidase (PDB ID: 3WY3) protein structure was downloaded from the PDB database; Pymol2.3.0 was used to remove protein crystalline water, primitive ligands, etc. The small molecule and protein structure were imported and saved into Auto-Docktools (v1.5.6) for hydrogenation, charge calculation, charge assignment, and specify atom type. Docking results were analysed for interaction patterns using PyMOL 2.3.0 and LIGPLOT V 2.2.4.”in line 183-189.
- On table 1 add a note on acarbose that it is a positive control compared to polyphenol.
Response: According to the reviewer’s suggestion, we have added the “positive control” in table 1.
- Table 3 to reorganize, as long as to decrease the font.
Response: According to the reviewer’s suggestion, we have decreased the font in table 3.
- Add reference at the beginning of the line 303.
Response: Thank you so much for your valuable comments. We have added the reference in lines 364.
- Paragraph 3.7, replace ITC by isothermal titration calorimetry.
Response: Thank you so much for your valuable comments, the “ITC” has been revised to “isothermal titration calorimetry (ITC)” in line 382.
- Rephrase the protocol of docking experiment: download the α- glucosidase (PDB ID: 3WY3) protein structure from the PDB database; use Pymol2.3.0 to remove protein crystalline water, primitive ligands, etc. Import the small molecule and protein structure into AutoDocktools (v1.5.6) for hydrogenation, charge calculation, charge assignment, specify atom type and save. The α- glucosidase (PDB ID: 3WY3) structure was downloaded from the PDB……etc.
Response: Thank you so much for your valuable comments. We have improved the protocol of docking experiment to “The α-glucosidase (PDB ID: 3WY3) protein structure was downloaded from the PDB database; Pymol2.3.0 was used to remove protein crystalline water, primitive ligands, etc. The small molecule and protein structure were imported and saved into Auto-Docktools (v1.5.6) for hydrogenation, charge calculation, charge assignment, and specify atom type. Docking results were analysed for interaction patterns using PyMOL 2.3.0 and LIGPLOT V 2.2.4.” in lines 183-189.
- line 153: Compared
Response: Thank you so much for your valuable comments. the “compared” has been revised to “Compared” in line 208.
- line 195: rephrase “no SY and AD effects were found”.
Response: Thank you so much for your valuable comments. the “no SY and AD effects were found” has been revised to “The SY and AD effects were not observed” in line 251-252.
- line 196: The combination of QU with.
Response: Thank you so much for your valuable comments. the “the combination of QU with” has been revised to “The combination of QU with” in line 253.
- Correct unit weighting of binding constants in table 4.
Response: Thank you so much for your valuable comments. We have corrected the unit weighting of binding constants in table 4.
- lines 344-346, rephrase and correct punctuation.
Response: Thank you so much for your valuable comments. We have improved to “The binding of KA to α-glucosidase is similar to QU, QI and is exothermic (Fig. 4 E), ΔS was greater than 0 while ΔH was less than 0.” in lines 406-408.
Round 2
Reviewer 1 Report
Manuscript titled “In vitro inhibitory effects of polyphenols from Flos Sophorae Immaturus on α-glucosidase: Action mechanism, isothermal titration calorimetry and molecular docking analysis" reports various in vitro and in silico analyses aimed at determining the glucosidase-inhibition potential of Flos Sophorae Immaturus. The compounds and their combinations were studied in vitro, while docking analyses were also performed to further study their activities. This version of the manuscript was revised according to comments and suggestions made during an initial revision, those made by the present reviewer include:
1. Fixing a typo in the title (change “polyphenols of” to “of polyphenols”). The typo was corrected as suggested.
2. Clarifying the authors’ intended meaning when they say that rutin content is “reaching 20%”. The phrasing was changed to “with rutin content up to 20% of the total mass of FSI”, which is more descriptive and removes ambiguity.
3. Defining the abbreviation “PNPG”. The abbreviation was revised to “4-nitrophenyl β-D-glucopyranoside (PNPG)”.
4. Clarifying the origin of the glucosidase used in the present work. The authors have mentioned that it is from yeast.
5. Specifying the concentration of phenolic used in sections 2.3-2.6, since only volumes added are stated, but not their concentrations. Concentrations used have now been specified as suggested.
6. Specifying the meaning of “brassin”. The typo was fixed.
7. Clarifying the “key role” mentioned for quercetin when present at low concentrations, and “low concentration FSI extracts”. The sentence was rephrased to “Quercetin played a key role in the α-glucosidase inhibition rate in FSI extracts, which was consistent with the IC50 results”, which eliminated the ambiguity of the original phrasing.
8. Homogenizing the number of decimal digits for data shown in tables. Decimal digits were homogenized as required in tables 1, 2 and 4.
9. Eliminating the unnecessary repetition of “0.02 mg/mL” shown in table 3. The repetitive data was eliminated and mentioned only once on the table’s footer.
10. Confirming data mentioned in “Supplementary 4”. “Supplementary 4” was revised to “Table 5”.
11. Clarifying the meaning of “(A)” when mentioning some amino acids. The “(A)” was eliminated.
12. Revising the manuscript to amend various typos and writing mistakes. The authors revised their document and corrected a number of minor details throughout.
According to the aforementioned changes, it is apparent that the authors considered and adequately addressed all comments made by the present reviewer. Thus, there are no additional ones to make for the current version of the manuscript.
Reviewer 2 Report
Author should only very some spell check in English